# Determinants of Young People with Secondary Education Being Employed

**Alina Stundziene ***  **and Vilda Giziene**

School of Economics and Business, Kaunas University of Technology, 44249 Kaunas, Lithuania
* Correspondence: alina.stundziene@ktu.lt

**Abstract:** The aim of this paper is to find out the main factors that determine whether young people with secondary education are employed or not in Lithuania. A survey of young people, aged 18–25, was carried out to gather information about individual characteristics and to find out the reasons why they are not employed. The analysis of the collected data was performed using independent samples tests and the calculation of the contingency coefficient. The research showed that young people start work quite young and are willing to enter the labor market. However, they find it difficult to combine work and study. The regression analysis found five significant variables to explain why young people are employed or not, i.e., their job contract, satisfaction with other work conditions, gender, the opportunity to work remotely, and 40 h worked per week. The probit model showed that temporary and full-time jobs reduce the probability of being employed; meanwhile, the opportunity to work remotely and greater satisfaction with other work conditions increases the probability of employment. The probit model also provided evidence that women are more likely to work than men.

**Keywords:** youth unemployment; inactivity; employment; labor market; education

---



## 1. Introduction

Youth unemployment is an economic and social problem in many countries. According to Eurostat (European Statistics. https://ec.europa.eu/info/departments/eurostat-european-statistics_en (accessed on 3 November 2022)), the unemployment rate was 6% in the EU-27 and 4.6% in Lithuania in September 2022. Meanwhile, the youth unemployment rate was 14.6% in the EU-27 and 11.6% in Lithuania in the same month. Youth unemployment for females is slightly higher (16.7% in the EU-27 and 14.7% in Lithuania in 2021) than for males (16.5% in the EU-27 and 14% in Lithuania in 2021). In general, the unemployment rate of people with tertiary education was 4.8% in the EU-27 and 3.8% in Lithuania in 2021. These statistics let us conclude that people with higher education can more easily enter the labor market.

Although the unemployment indicators in Lithuania are better than the average in the EU, the activity rate of young people there is quite low. The activity rate was 73.6% in the 15–64 age group and 39.3% in the 15–24 age group in the EU-27 in 2021. Meanwhile, it was 78.2% in the 15–64 age group and 36.3% in the 15–24 age group in Lithuania in the same year.

The causes of youth unemployment or inactivity differ in each country and specific solutions are needed to solve this problem. Abzhan et al. (2020) analyzed different data from EU countries and found that employers should also participate in the education system and cooperate to guarantee that young people are ready to join the labor market. Based on research, a mismatch between demand and supply exists, and a lack of skills and knowledge are the main reasons of youth unemployment (Manacorda and Petrongolo 1999; Assaad and Roudi-Fahimi 2007; Quintini 2014; Ashton et al. 2005; Iversen and Stephens 2008; Maurer 2011; Dalziel 2017; and Valiente et al. 2020). This can be explained by the malfunction of the education and training system and/or low investment in individual

education and training (McQuaid and Lindsay 2005). It is important to ensure adequate and high-quality study and learning throughout life in order to reduce unemployment. Governments should make the education and training system more responsive to the needs of countries' economies (Almeida et al. 2012).

Young people are the most important members of the future labor market. A high level of youth unemployment has a negative impact on the individual and the whole country (Gordon 2010; and Putun et al. 2017). From the point of view of the individual, unemployment reduces the income of individuals and causes psychological and social problems (Pohlan 2019). From the point of view of the state, it deteriorates its financial situation and slows down the economy. If the problem of youth unemployment is not solved, the country suffers from the aging of the workforce and a decline in labor productivity. Young people's participation in the labor market is very important for faster economic development.

Young people make up an important share of the labor market in Lithuania. Their ability and attitude to enter the labor market can have a significant impact on their future employment and even the whole labor market. Therefore, it is important to analyze the primary work experience, satisfaction, and expectations of young people. Although the young's people aged 25–29 employment rate r in Lithuania is one of the highest in Europe, it is close to the EU average for people aged 20–24 and lower than the EU average for people aged 15–19. As many young people aged 15–24 still study, it is useful to find out the reasons why they are employed or inactive and their experiences of combining their studies and work.

Some research has been carried out on youth unemployment in Lithuania, but each study takes different aspects into account. Braziene (2020) found that parents and parental education have an impact on the participation of young people aged 16 to 29 years in the labor market. Based on the experience of Lithuanian university graduates, Kvedaraite (2014) analyzed entrepreneurship as a tool for youth participation in the labor market. Rauckienė-Michaelsson and Acienė (2019) revealed the social portrait of young people in Lithuania from the perspective of youth policy, assessing aspects such as unemployment, emigration, attitudes towards the family, opportunities for social participation in society, and the development of solidarity. Repečkienė et al. (2012) analyzed the dynamics of youth unemployment, aiming to find measures that could facilitate the transition of academic young people from educational institutions to the labor market and thus reduce their unemployment rates.

The aim of this research is to find out the main factors that determine whether young people with secondary education are employed or not in Lithuania. People who do not have secondary education were not analyzed in this research, as this is a specific group, which is characterized primarily by a lack of qualifications, a lack of motivation to work, and individuals who are poorly prepared for the labor market. In general, the success or failure of young people in the labor market is influenced by a combination of individual characteristics and by the situation in the labor market. This research is limited to the analysis of individual characteristics, such as gender, age, educational status, work experience, etc. These aspects have not been analyzed in the previous research based on Lithuanian data.

## 2. Literature Review

According to the OECD, the youth unemployment rate is represented by the number of unemployed 15–24-year-olds expressed as a percentage of the youth labor force (overall number of both employed and unemployed youth). Young people are more vulnerable than adults, and it is harder for them to enter the labor market. This is influenced by a lack of work experience, educational attainment, and difficulty in making the transition from studies to the labor market. Unemployment costs are felt not only by individuals but also by society (O'Higgins 2015).

Kang (2021) analyzed the causes of youth unemployment in OECD countries and EU member states in the 2000–2017 period and found that not only business cycles but also

other factors such as temporary contracts, education, and a lack of dual education affect youth unemployment. Papik et al. (2022) carried out research in Slovakia (he analyzed 464 Slovak high schools from the National Institute for Certified Educational) and found that youth unemployment is closely linked to a country's education system. They found that the state should improve the education system by including a dual learning system and the possibility of part-time work for students. Closer cooperation between schools, universities, and employers would lead to an easier transition from the education system to the labor market (Succi and Canovi 2019). Education provides knowledge and skills, and graduates are more likely to enter the labor market (Korpi et al. 2003). Location also has an impact on youth unemployment. It is easier to find a job in a big city than in a suburb or small town (Ciburiene 2016). The analysis of empirical studies identifies four main factors that contribute to youth unemployment:

- Business cycles. Youth unemployment is more responsive to business cycles due to the fact that young people have less work experience, a lack of education, and are employed in less skilled jobs (Brada et al. 2014; Condratov 2014; Bruno et al. 2016; Kang 2021; and Tomić 2018).
- Demographic, individual, and social conditions, i.e., migration, barriers to regional mobility, changes in the structure of the economy, and a skills mismatch in the labor market (Korenman and Neumark 2000; and King and Williams 2017).
- Policies and policy instruments, i.e., labor market regulation. Youth unemployment is influenced by work-related taxes, unemployment benefits, the minimum wage (Bernal-Verdugo et al. 2012), temporary employment contracts, and measures to integrate young people into the labor market.
- Education, that is, a poor education system (Pastore 2019).

Hofferth and Collins (2000) argued that all individuals need to participate in the labor market, especially young people. Youth participation in the labor market increases the flexibility of the labor market and improves the quality of life of the individuals themselves, and in a broad sense, it may mean an improvement in working conditions.

However, young people often have expectations that are too high and exceed their potential. Šafránková and Šikýř (2017), who carried out research in the Czech Republic, discovered that young people have very high expectations that do not match their qualifications and experience, e.g., they want to earn a high income, move up the career ladder quickly, etc. O'Reilly et al. (2015) highlighted that youth unemployment in the EU is also driven by family status and family experience (if the parents have been out of work for a long time, it is also more difficult for the children to find jobs). We need a holistic approach to understanding youth unemployment that should include flexibility in the labor market and family experience.

A key factor in reducing unemployment is investment in human capital, that is, investment in education, qualifications, knowledge, and skills (McMahon 1997; Wolfe and Zuvekas 1997; and Vila 2000). The main cause of youth unemployment is a lack of necessary knowledge and skills (Dietrich and Möller 2015). European countries solve the problem of youth unemployment differently. Dietrich and Möller (2015) state that there should be a united system to deal with this problem. Choosing studies according to the demand in the labor market would ensure a reduction in youth unemployment. Parents in Italy have a great influence on young people and should provide them with appropriate guidance and advice (Giannelli and Monfardini 2003).

Young people spend a lot of time studying, so they often have temporary jobs. Usually, they cannot find permanent jobs and have poorer working conditions than other employees (Papoutsaki et al. 2019). They are rarely self-employed. They are often employed in the service sector, where work is often physical and routine.

Education has a significant impact on youth unemployment (Fergusson and Yeates 2021) in terms of access to employment and in terms of wage differentials. Education can act as a buffer against unemployment in times of economic downturn (Scandurra et al. 2021). Although graduates have greater employability, a university degree does not

always guarantee that an individual will acquire a job quickly or acquire the job they want (Carter 2019). Higher education can widen the gap between what employers need and the knowledge and skills acquired through study. Young people lack informal and intrinsic human capital, which is acquired through work and vocational training, making it difficult for them to acquire a job and leading to high rates of youth unemployment (Refrigeri and Aleandri 2013).

Hedvicakova (2018) aimed to find out how young people managed to find their first job, how they searched for it, what barriers they faced, what experience and knowledge they brought to the labor market, and what their future plans were. A survey of graduates from the Czech Republic University of Hradec Kralove revealed that the majority of graduates are employed and only a small number are setting up their own businesses. The author concluded that the education system and the labor market need to work together and that it is necessary to train a broader range of professionals. Communication skills, information and computer skills, language skills, problem-solving, decision-making, responsibility, flexibility, teamwork, and leadership skills are important. Too high expectations are another problem of youth unemployment. They wish to acquire a job close to home, to obtain a high salary, and to acquire a job that is relevant to one's specialization. The author argues that lifelong learning must be encouraged.

Universities should develop students' ability to solve real-world problems, develop cognitive skills, and learn to work in teams (Succi 2019). Succi and Canovi (2019) state that it is not enough to acquire specialized knowledge, but soft skills are also very important, which is a scarcity today, and this is the reason why graduates have problems entering the labor market.

However, obtaining higher education is not a guarantee of a successful pathway into the labor market (de Weert 2007). Research in European countries concludes that graduation is only one step to entering the labor market, and people need to study and increase their knowledge during their whole life (Carter 2019). Graduates face multiple challenges. They accept lower positions than they would like or a job that is not in line with their competencies (Clarke 2017). Higher education does not necessarily close the gap between employers' needs and the knowledge and skills acquired through studies. Young people lack informal and intrinsic human capital, which is acquired through work and vocational training, making it difficult for them to acquire a job and leading to high rates of youth unemployment (Refrigeri and Aleandri 2013).

Ayllón et al. (2021) analyzed youth unemployment in 12 European countries and argues that long-term unemployment is a signal to employers that an individual's human capital is diminished and that the individual is less motivated or not productive. The longer an individual is out of work, the harder it is to find a job, and the more likely it is that, if he or she does find a job, it will be at a low wage level, which has a negative impact on the economy as a whole.

According to McQuaid (2015), unemployment at the beginning of a career can lead to poorer cognitive skills, such as the ability to process information and learn. This can affect a young person's adaptability, productivity, expected salary, and ability to find suitable work later in life.

The analysis of the scientific literature let us conclude that youth unemployment depends not only on economic and political factors, but also on educational attainment, accumulated knowledge, personal adaptability, and the willingness to develop and learn throughout life. From a state perspective, there should be opportunities for closer cooperation between higher education institutions and businesses, greater responsiveness to the needs of the labor market, and opportunities for young people to work part-time while studying. In European countries with dual training, young people enter the labor market more easily than in those without dual training.

## 3. Methodology

A survey of young people, aged 18–25, was carried out to gather information on individual characteristics and to find the reasons why they are not employed. It was carried out online. The survey link was sent to high school students and graduates. They were also asked to share this link with their friends aged 18–25 who are studying or have higher education. The questionnaire was open from January 2022 until October 2022. It covered questions about gender, age, studies, and their experience in the labor market.

Two hundred and forty-three well-filled questionnaires were received at the end of October. According to Statistics Lithuania data (https://osp.stat.gov.lt (accessed on 1 October 2022)), there are approximately 222,420 young people, aged 18–25, in Lithuania, but approximately 5.4% do not have secondary education and do not study anywhere. Choosing a 95% confidence interval and a 50% sample proportion, which produces the maximum possible variation, the sampling error is 6.3%.

The analysis of collected data was performed using several independent samples tests:

- The Mann–Whitney U test was used to compare differences between two independent groups, that is, employed and unemployed. It tests the equality of two distributions and is used when the dependent variable is ordinal or continuous but not normally distributed.
- The Kruskal–Wallis H test is an extension of the Mann–Whitney U test. It is the non-parametric analog of one-way analysis of variance and detects differences in distribution location when there are more than two samples.
- The Kolmogorov–Smirnov test is used to test whether the maximum absolute difference in the overall distribution of the two groups is significant.
- The median test is a more general test and also detects distributional differences in location and shape.

The purpose of these tests is to detect the determinants that distinguish young people between employed and unemployed. However, these tests are used only if the indicators are continuous or ordinal. In the case of nominal data, the contingency coefficient is calculated. It is a symmetric measure that is based on the chi-square statistic. The contingency coefficient takes values between 0 and $\sqrt{(k-1)/k}$, where $k$ is the number of rows or columns, whichever is smaller.

When the most important characteristics are found, the binary probit model is estimated:

$$P(y = 1/x_i \ldots x_k) = b_0 + b_j x_j + \ldots + b_k x_k \tag{1}$$

Here, $y$ is employment status, $x_i$, $i = 1, \ldots, k$, is the other characteristics of the respondents, and $b_j$, $j = 0, 1, \ldots, k$, is the parameters of the model. The probit model is used to estimate the effects of one or more independent variables on a dichotomous dependent variable. The dependent variable in this research is set to 1 if a respondent currently works and to 0 if a respondent does not work.

The Hosmer and Lemeshow (1989) test is performed as a goodness-of-fit test. It determines whether the model adequately describes the data. The Hosmer–Lemeshow statistic indicates a poor fit if the $p$-value is less than 0.05. It compares the fitted expected values to the actual values by group. If these differences are large, the model is rejected as providing an insufficient fit to the data.

## 4. Results and Discussion

Two hundred and forty-three respondents aged 18–25 participated in the survey. Of the respondents, 64.2% of them were men and 82.7% of the respondents study in a high school, 9.9% already have higher education, and 18.1% have post-secondary non-tertiary education. Forty-eight percent of the respondents currently work, while 16% have never worked before (Figure 1).

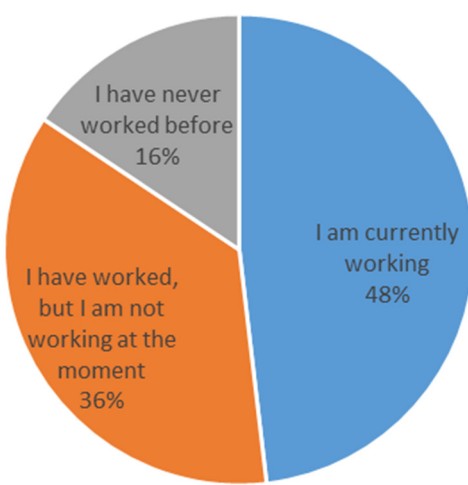

**Figure 1.** The status of employment.

Young people were asked to grade 11 criteria, mentioned in Figure 2, according to their importance when choosing a job, indicating a grade from 0 (not important at all) to 5 (very important). As indicated by the respondents, the main criterion when looking for a job is the type of job. Sixty-five percent of the respondents said they must like it (65% graded it by 5, i.e., very important). The salary, opportunity to improve, and career opportunities, as well as the possibility of combining work and studies, are also among the most important criteria (Figure 2). The main reasons for leaving a job are also studies, dissatisfaction with the job type, and salary.

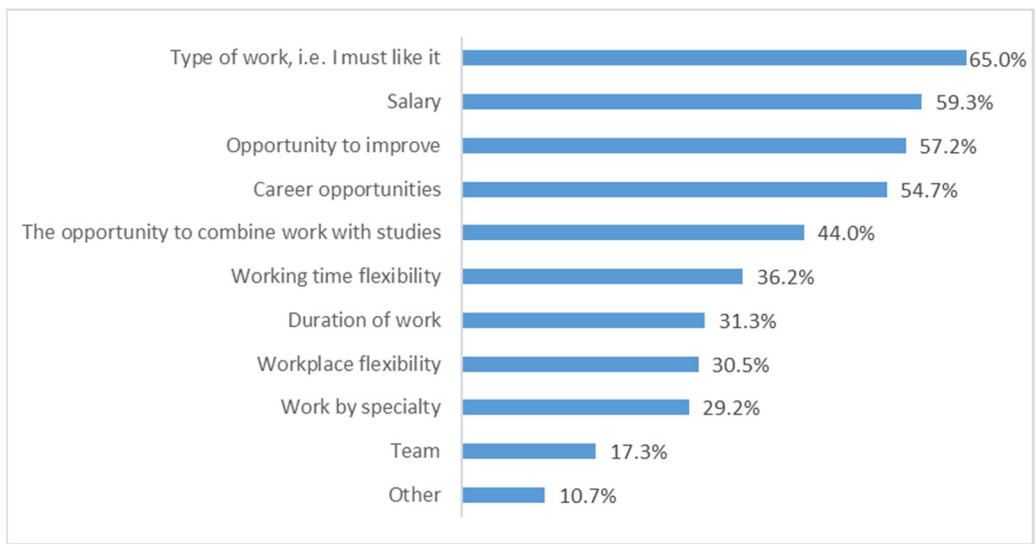

**Figure 2.** The most important criteria for choosing a job.

Young people were also asked to indicate the reasons why they are unemployed or inactive, choosing one or more answers from the list given, presented in Figure 3. The results show that 41.3% of the respondents are inactive due to their studies and 44.5% of students would like to work but are unable to find a job with a flexible work schedule or one that could be combined with their studies. Almost 24% of the respondents are trying to find a job in their field (specialty) but have failed. However, 31.8% of the respondents do not see the need to work, as they are supported by other people (e.g., parents) or benefits received from the employment service, municipality, or other institution and they are sufficient (Figure 3).

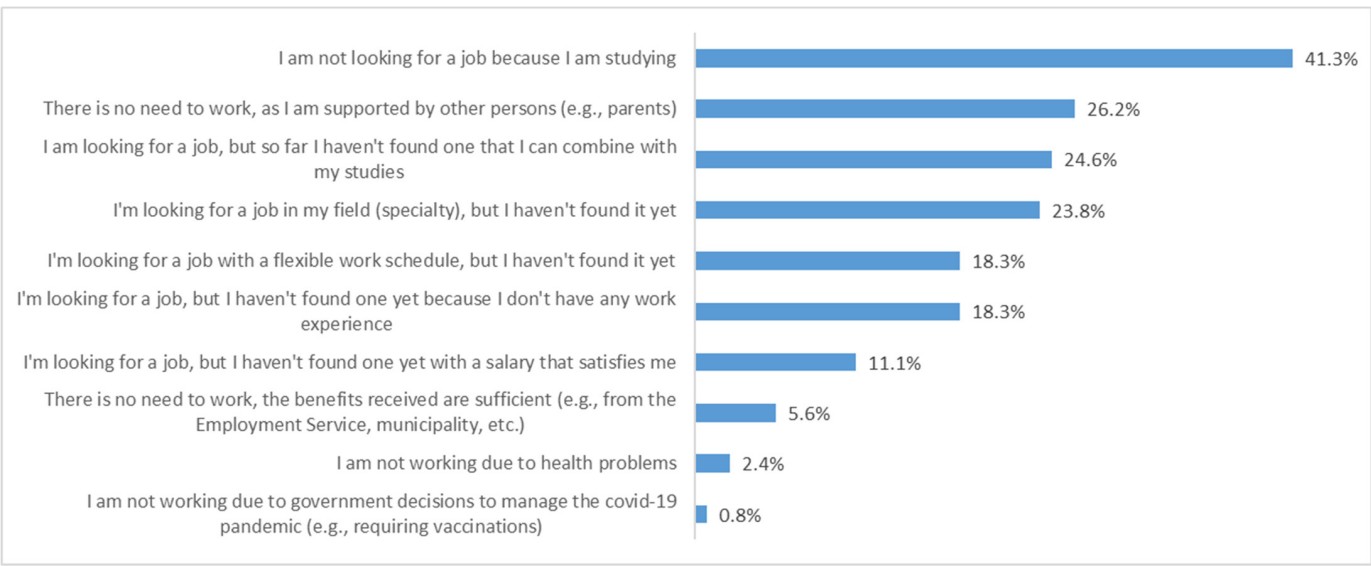

**Figure 3.** Reasons for unemployment or inactivity.

### 4.1. Differences between Working and Non-Working Young People

The further analysis is based on the comparison of young people who are currently employed and who are currently not employed in order to clarify the reasons for it.

Most of the respondents started to work when they were 18 or even 16. Half of the respondents started their first job when they were 18 years or younger. The median age and its distribution do not differ significantly between the respondents who currently work and those who do not work currently but worked earlier (Table 1). Their first job is usually temporary and unqualified, most likely during the summer holidays (Figure 4). Working according to their specialty is only relevant for 16% and having a permanent job is only typical for 14% of respondents. That is why their first job was found to be quite easy. From the results, 84% of the respondents found it within one month and the search for it lasted for less than 3 months for 14% of the respondents.

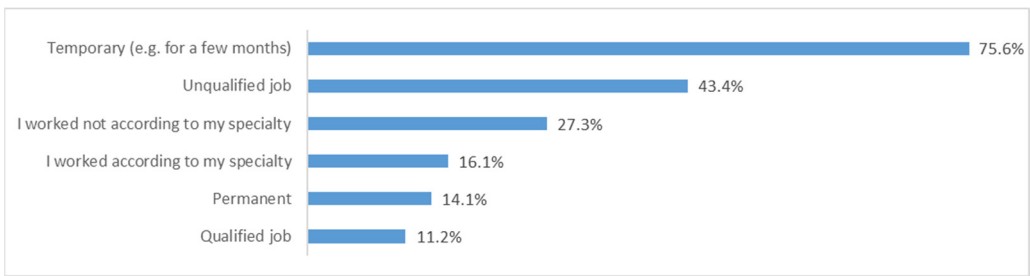

**Figure 4.** Type of first job.

Talking about their current job or the last job that the respondents worked, it was permanent (under an open-ended employment contract) for 87.7% of respondents who currently work, and the last job that they worked was temporary (under a fixed-term employment contract) for 70.5% of respondents who do not work currently. Therefore, there is a fairly strong relationship between employment status and the type of work. The contingency coefficient between these indicators is 0.512 and is significant (the *p*-value is 0.000).

**Table 1.** Results of independent samples tests (*p*-values).

| Indicator | Independent-Samples Median Test | Independent-Samples Mann–Whitney U Test | Independent-Samples Kolmogorov–Smirnov Test | Independent-Samples Kruskal–Wallis Test |
|---|---|---|---|---|
| Age | 0.004 | - | - | 0.001 |
| Age of respondents when they started their first job | 0.132 | 0.496 | 0.624 | - |
| Number of jobs | 0.732 | 0.097 | 0.100 | - |
| Work experience | 0.003 | 0.000 | 0.007 | - |
| Satisfaction with their first job | 0.997 | 0.641 | 0.823 | - |
| Satisfaction with the type of the last job | 0.000 | 0.000 | 0.000 | - |
| Satisfaction with the salary of the last job | 0.135 | 0.001 | 0.004 | - |
| Satisfaction with the working hours in the last job | 0.000 | 0.000 | 0.000 | - |
| Satisfaction with other work conditions in the last job | 0.000 | 0.000 | 0.000 | - |
| Salary | 0.049 | 0.071 | 0.223 | - |

Type of work is also related to the required qualifications. It is an unqualified job for at least 67.1% of respondents who work temporarily and it is a qualified job for at least 59.5% of respondents who have a permanent job (some respondents could not specify if it was a qualified or unqualified job). Almost half of the respondents (47.3%) who are currently employed work according to their specialty, and for only 14.9% of the respondents who do not currently work, their last job was according to their specialty.

The average number of companies they have already worked for is almost three and does not significantly differ between young people who currently work and those who are not working but worked earlier. However, the spread of the number of jobs of those respondents who do not work currently is slightly wider (Figure 5). Meanwhile, work experience is significantly different in these two groups of respondents (Table 1). The median value is two years and one year for respondents who currently work and do not work, respectively.

Their satisfaction with their first job is rated as average (3 out of 5) and it also does not differ significantly between these two groups of young people (Table 1). Satisfaction with their current job for those who currently work is one point higher (4 out of 5), but there is no relationship between the evaluation of the first and the last job. The Spearman rank correlation coefficient between their satisfaction with their first and their last job is 0.057 and is not significant (the *p*-value is 0.417). However, their satisfaction with their current job or their last job if a person is not employed currently differs between the two employment groups. The median evaluation of working hours is approximately 5 and 4 in the group of currently working and non-working young people, respectively. Salary and other working conditions are evaluated as 4 and 3, respectively. The difference in the median evaluation of the type of work is the largest, that is, 4.5 and 3, respectively.

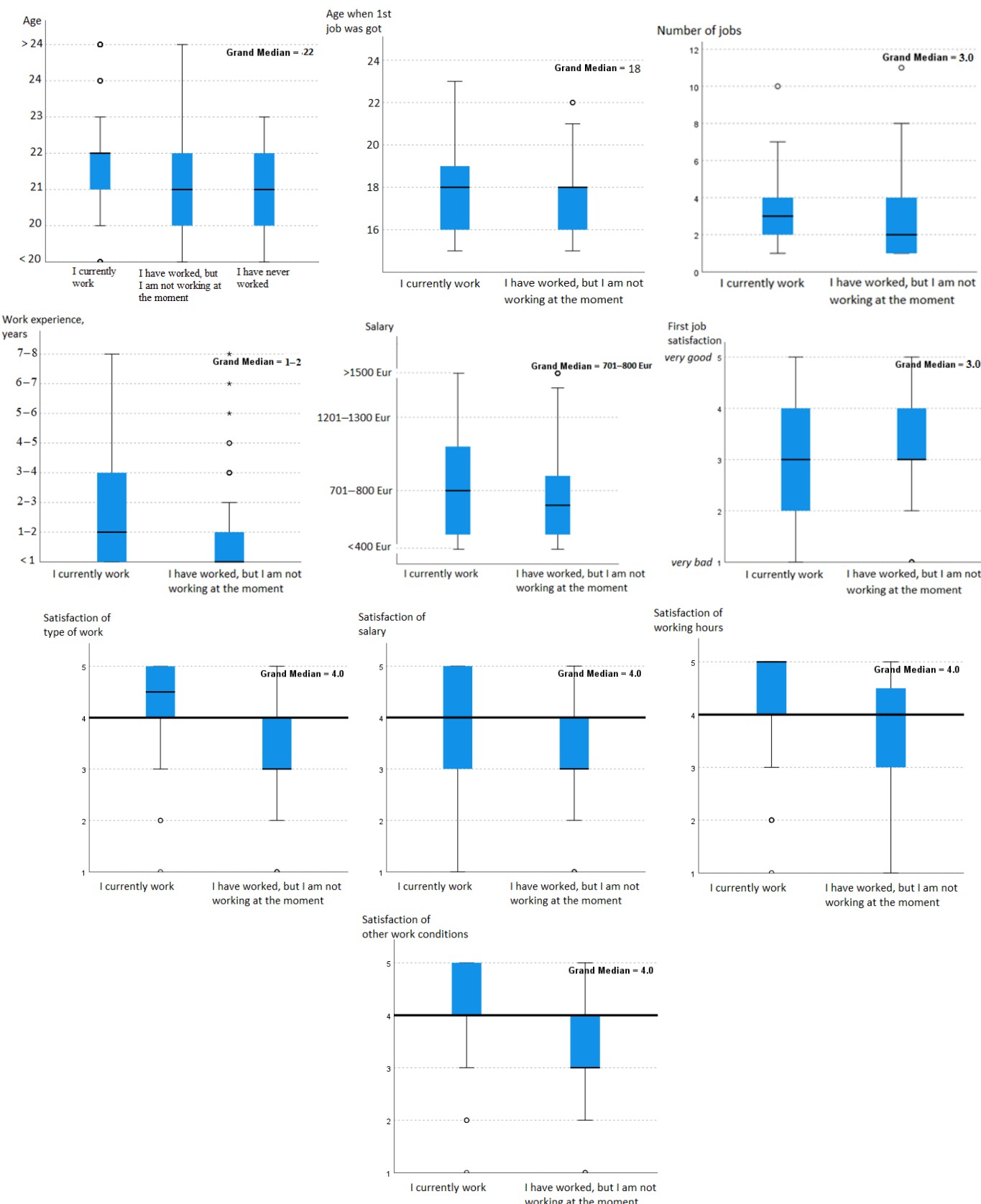

**Figure 5.** Boxplots drawn based on the independent samples' median test. Note: circle represents an outlier and asterisks (*) represents an extreme outlier.

Approximately 66.7% of the currently employed young people had the possibility to work remotely during the COVID-19 pandemic, while just 19.5% of the currently unemployed people had such a possibility in their last job. The contingency coefficient between the possibility of working remotely and the employment status is 0.424 and is significant (the *p*-value is 0.000). Of the respondents, 84.3% of currently employed young people and 50.5% of currently not employed people have the possibility to work with a flexible work schedule. The contingency coefficient between the ability to work flexibly and employment status is 0.342 and is significant (the *p*-value is 0.000).

Women are more likely to work than men (Figure 6). Approximately 40% of men work, and the others do not work currently or have never worked. Meanwhile, 62% of women work and only 12% have never worked. The contingency coefficient between gender and employment status is 0.212 and is significant (the *p*-value is 0.004).

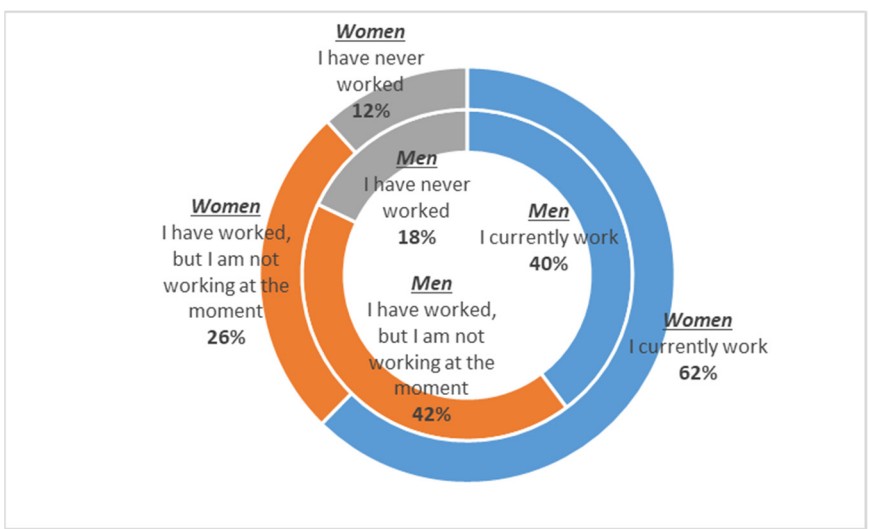

**Figure 6.** Employment status by gender.

Almost half (46.2%) of those who currently work are 22 years old. Meanwhile, more than half (56.3%) of those currently unemployed are under 22 years of age (Figure 5). The median and age distribution significantly differ among the employment status groups (Table 1). The contingency coefficient between age and employment status is 0.305 and is significant (the *p*-value is 0.016).

The study place is considered taking into account the following answers: I am a student in a post-secondary non-tertiary school, I am a student in a college, I am an undergraduate student in a university, I am a post-graduate student in a university, and I do not study. Undergraduate students account for the largest group in all of the categories of employment status, that is, they account for 73.5% in the group of respondents who work, 88.6% in the group of respondents who do not work currently, and 97.4% in the group of respondents who have never worked. However, the correlation analysis found a significant relationship between employment status and the study place. The contingency coefficient is 0.280 and is significant (the *p*-value is 0.008).

Young people who currently do not work used to work longer hours in their last company compared to young people who currently work (Figure 7). The results show that 45.6% of the respondents who are currently not employed work 40 h per week, 14.1% of the respondents work more than 40 h per week, and 14.1% of the respondents were unable to specify the number of normal working hours because the number of working hours varies during the week. Meanwhile, 42.6% of the currently employed respondents work 40 h a week (full-time), 21.7% of the respondents work 20 h (part-time), and 11.3% of the respondents work 30–39 h a week. The correlation analysis found a significant relationship

between the hours worked and employment status. The contingency coefficient is 0.283 and is significant (the *p*-value is 0.008).

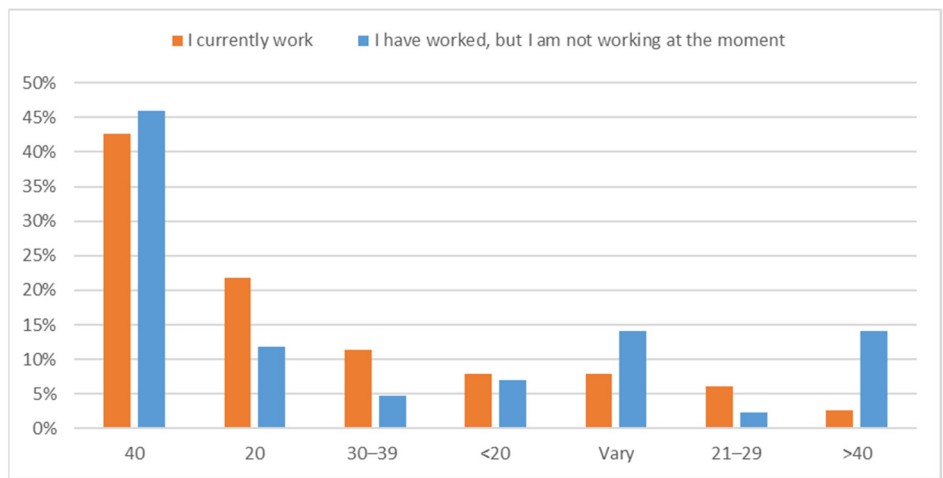

**Figure 7.** Hours worked per week.

Meanwhile, the desired duration of work does not differ significantly between the two groups of respondents. The results show that 37.2% of the respondents who work and 34.9% of the respondents who do not work currently would like to work 40 h per week (full-time). A figure of 27.4% and 26.7% of respondents would like to work 30–39 h, and 23.9% and 19.8% would like to work 20 h per week (part-time), respectively. Thus, the decision to stop working can be made based on the long working hours experienced.

Although the median salary in the two groups of young people differs significantly (at a significance level of 0.05), the distributions do not differ significantly. The median salary of the respondents currently employed is EUR 701–800 and it was EUR 601–700 in the last job of the currently unemployed. The minimum net wage in 2022 was EUR 518, but 28.9% of the currently employed and 32.9% of the currently not working people received less than EUR 500 per month.

*4.2. Probit Model*

The previous analysis shows that the following criteria are significant in dividing the respondents into employed and unemployed: gender, age, study place, hours worked, job contract (temporary or permanent), qualifications needed for a job, working according to specialty, last job satisfaction (based on its type, salary, working hours, and other conditions), and the opportunity to work remotely and flexibly, as well as work experience. All of these factors were used in a further analysis to create a probit model in order to predict employment status. As most of these indicators are qualitative, dummies were created. As 26 independent variables were under investigation, a stepwise forward procedure was performed first in order to find out the most important indicators. Five significant variables were found, i.e., job contract (1 is a temporary job and 0 is a permanent job), satisfaction with other work conditions (ranged from 1 to 5), gender (1 is a man and 0 is a woman), the opportunity to work remotely (1 if there is an opportunity to work remotely and 0 if there is not), and 40 h worked per week (1 is if it is a full-time job and 0 is in any other case). These variables are used to create a probit model. The dependent variable is set to 1 if a respondent works currently and 0 if a respondent does not work, i.e., is unemployed or inactive currently, but worked earlier and his/her previous work experience is considered. The results (marginal effects) of the probit model are presented in Table 2.

**Table 2.** Estimates of the probit model.

| Variable | Coefficient | Std. Error | z-Statistic | Prob. |
|---|---|---|---|---|
| C | −1.3658 | 0.5250 | −2.6017 | 0.0093 |
| Job contract | −1.7276 | 0.2620 | −6.5930 | 0.0000 |
| Satisfaction with other work conditions | 0.6260 | 0.1333 | 4.6962 | 0.0000 |
| Gender | −0.5949 | 0.2536 | −2.3456 | 0.0190 |
| Opportunity to work remotely | 0.9156 | 0.2571 | 3.5615 | 0.0004 |
| 40 h worked per week | −0.6003 | 0.2641 | −2.2729 | 0.0230 |
| McFadden R-squared | | 0.4985 | | |
| LR statistic | | 135.6789 | | |
| Prob(LR statistic) | | 0.0000 | | |

All five independent variables are significant, and the model is significant as well. McFadden's R-square is nearly 0.5. The results show that temporary and full-time jobs reduce the probability of being employed. Approximately 70.5% of the respondents who do not currently work had a temporary job earlier on, and 87.7% of the respondents who currently work have a permanent job. Permanent job contracts increase the probability of being employed. Most young people try to keep their job if they acquire one. Only 20.6% of the respondents who indicated having a permanent job contract are not working currently, and the inability to combine work and studies is the main reason for leaving it. Almost half of the respondents indicated that they would like to work but cannot find a job with a flexible work schedule or one that could be combined with their studies.

Meanwhile, the opportunity to work remotely and greater satisfaction with other work conditions increase employment probability. Other work conditions include the work environment, not including the type of work, the salary, and the schedule. Those who currently work are more satisfied with the other work conditions than those who do not work but worked earlier. The mean satisfaction level is 3.4 for other work conditions in their previous job (for those who currently do not work) compared to 4.3 for the mean satisfaction in their current work (those who currently work). Approximately 65.0% of the respondents who currently work have the opportunity to work remotely, and approximately 86.5% of the respondents who do not currently work had no possibility to work remotely in their previous job. The probit model also proves that men have a lower probability of being employed than women. As indicated by 63.1% of women and only 39.7% of men currently working. This shows that women are more capable of combining work and studies than men. This also correlates with the attitude of Lithuanians that women can cope with more work than men.

Moreover, higher rates of employment and activity for females compared to males could be a new trend in Lithuania. According to the data from Statistics Lithuania, the historical activity rate of young males is approximately 7 percentage points higher than that of young females (aged 15–24). However, recently, this gap has narrowed, and the activity rate of females was 3.2 percentage points higher than the activity rate of males in the third quarter of 2022 (Figure 8). The employment rate of males usually outperforms the employment rate of females. However, it manifests itself when the economy is growing. When the economy slows or declines, the employment rate of young males declines faster than the employment rate of young females and even becomes lower than the latter. This was seen during the financial crisis in 2009 and during the COVID-19 pandemic. The employment rate of young females was almost 3 percentage points higher than the employment rate of young males in the third quarter of 2022.

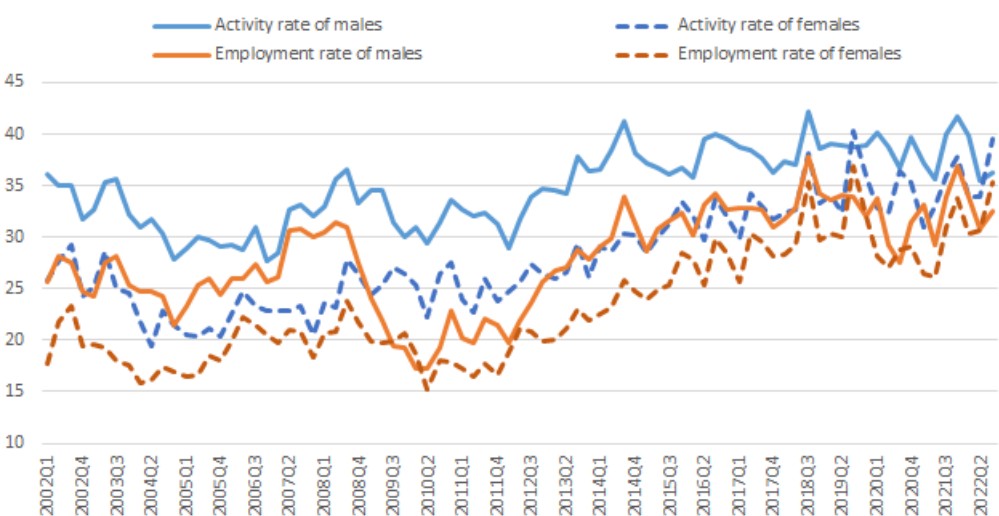

**Figure 8.** Activity and employment rates of males and females, aged 15–24, in Lithuania.

There have been several studies that have also found similar results with respect to the gender aspect in the labor market. However, there is a lack of studies that analyze the differences in the employment rates of young males and females with secondary education. We failed to find studies that incorporate all three of these indicators, i.e., gender, age, and education. Based on panel data from 41 African countries, Baah-Boateng (2016) found higher youth employment rates among females than males in general. Meanwhile, Cipollone et al. (2014), who carried out an analysis of trends in women's participation in the labor market over a 20-year period in 15 EU countries, state that trends in female participation in the labor market vary considerably across countries. It depends on the country's social policy, laws, and whether a woman has children or not. Women's ability to remain in the labor market depends on their ability to combine work and household responsibilities (Koutentakis 2015). Such compatibility could be achieved through flexible working arrangements, such as part-time work. Conde-Ruiz and Marra de Artinano (2016) state that the difference between male and female unemployment in Italy is due to the inability to work part-time and over-qualification requirements, as well as the unequal distribution of tasks within households. Bičáková (2016) adds that gender differences in unemployment across countries are primarily driven by differences in women's post-parental labor market participation behavior, i.e., the duration of family leave and women's later entry into the labor market. The state should adopt policies that facilitate their return to the labor market after childbirth.

Table 3 presents correct and incorrect classification tables based on the prediction cut-off value of 0.05. Of the respondents, 84.9% of the unemployed respondents and 87.6% of the employed respondents are correctly classified by the estimated model. In general, the estimated model correctly predicts 86.4% of the observations. The gain in the number of correct predictions provides a measure of the predictive capacity of a model. The estimated model improves on the Dep = 0 (i.e., the respondent is not employed) predictions by 84.9 percentage points, but does more poorly on the Dep = 1 (i.e., the respondent is employed) predictions (−12.39 percentage points). In general, the estimated equation is 29.7 percentage points better at predicting responses than the constant probability model. This change represents a 68.6% improvement over the 43.2% correct prediction of the default model.

**Table 3.** Expectation prediction evaluation for binary specification.

| | Estimated Equation | | | Constant Probability | | |
|---|---|---|---|---|---|---|
| | **Dep = 0** | **Dep = 1** | **Total** | **Dep = 0** | **Dep = 1** | **Total** |
| % Correct | 84.88 | 87.61 | 86.43 | 0.00 | 100.00 | 56.78 |
| % Incorrect | 15.12 | 12.39 | 13.57 | 100.00 | 0.00 | 43.22 |
| Total Gain | 84.88 | −12.39 | 29.65 | | | |
| Percent Gain | 84.88 | - | 68.60 | | | |

The calculated Hosmer–Lemeshow statistic is 4.88 and the probability of $\chi^2(8)$ is 0.77, indicating that there are no large differences between the actual and predicted values.

**5. Conclusions**

Young people start working quite young and are willing to try their hand in the labor market. However, it is difficult for them to combine work and study. That is why most young people work only during the summer holidays. Because of this, most young people work in unqualified jobs. However, young people are quite flexible, adaptable, and not very demanding of temporary work. Most of the respondents rate their satisfaction with their job as average or higher. They do not need much time to find a job and change it quite often.

The following criteria have been found to be important when dividing the respondents into employed and unemployed: gender, age, study place, hours worked, job contract (temporary or permanent), qualifications needed for a job, working according to their specialty, last job satisfaction (based on its type, salary, working hours, and other conditions), and the opportunity to work remotely and flexibly, as well as work experience. However, the regression analysis found only five variables significant, that is, the job contract, satisfaction with other work conditions, sex, the opportunity to work remotely, and 40 h worked per week. The probit model showed that a temporary and full-time job reduces the probability of employment, meanwhile, the opportunity to work remotely and greater satisfaction with other work conditions increases the employment probability. The probit model also provides evidence that women are more likely to work than men.

Thus, the crucial thought is to ensure the possibility for young people who are studying to combine their studies with a job, providing the possibility to work remotely and flexibly, as well as fewer hours. The analysis showed that young people who are not currently employed used to work longer in their last company compared to young people who are currently working, but the desired duration of work does not differ significantly between the two groups of respondents. Therefore, the decision to stop working can be made based on the long working hours experienced. The type of job (they must like it), salary, the opportunity to improve, and career opportunities are also important for young people.

In summary, the reasons for being inactive are ambiguous, that is, voluntarily and involuntarily. Based on this research, nearly one-third of the respondents did not see the need to work (voluntary choice of being inactive). However, almost half of the respondents would like to work but are unable to find a job with a flexible work schedule or one that could be combined with their studies. It is complicated to study and work at the same time in Lithuania, since bachelor's degree studies take place during the day and attendance at most lectures is mandatory. Meanwhile, many businesses offer daytime jobs as well. Therefore, if certain measures were taken, the problem of inactivity or unemployment would be reduced. The results of this research are in line with the results of some other research, e.g., the results of Papoutsaki et al. (2019), who found that young workers in the UK are much more likely to be in a temporary job because they cannot find a permanent one, or the results of Papik et al. (2022), who suggested making it possible for Slovak students to work part-time. We recommend for the Lithuanian government to consider regulations on more flexible working conditions or even for them to give incentives to companies that

provide young people with the possibility to work and study. Closer cooperation between employers and high schools is needed to meet the needs of the labor market.

**Author Contributions:** Conceptualization, A.S. and V.G.; methodology, A.S.; software, A.S.; validation, A.S. and V.G.; formal analysis, A.S. and V.G.; investigation, A.S.; resources, A.S.; data curation, A.S.; writing—original draft preparation, A.S. and V.G.; writing—review and editing, A.S. and V.G.; visualization, A.S. and V.G. All authors have read and agreed to the published version of the manuscript.

**Funding:** This research received no external funding.

**Informed Consent Statement:** Not applicable.

**Data Availability Statement:** Not applicable.

**Conflicts of Interest:** The authors declare no conflict of interest.

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
