# Peer review of "Determinants of Young People with Secondary Education Being Employed"

_economies, doi:10.3390/economies11020040_

Round 1
Reviewer 1 Report
Introduction – the section that discusses the labor market for your people in Lithuania would benefit from statistics and data. For example, what is the unemployment rate for your people? What is the labor force participation rate? How does this compare to other countries in the region?
The literature review needs to be rewritten with a focus on clarity and organization. As is the literature review is redundant, at least one sentence is repeated word for word. The literature review also needs to go into more depth on the studies it cites. What is the geographical location of these studies? What was their sample and methodology? The literature review would also benefit from a concluding paragraph that summarizes previous findings and outlines how the current study contributes to the literature.
Methodology – much more information is needed on how the survey was conducted. For example, how was the survey advertised? Were any incentives offered for completing the survey? Do individuals without access to the internet have an opportunity to complete the survey?
Results and discussion
Please add a summary statistics table including data on age, sex, educational status, working status….
Figure 2 needs more explanation. Please include the actual survey question. Was the question open ended or were respondents given these choices? Could they cite more than one reason?
Same comment for Figure 3 – you need more information on the exact question being asked.
The probit model is unclear and problematic. Wouldn’t only those that are employed have a job contract? How can someone without a job by satisfied or not by with other work conditions? Wouldn’t job satisfaction only be applicable to those with a job? What do you mean by “other work conditions?” 40 hours worked per week would obviously only apply to those with a job.
Table 2 should show marginal effects or odds ratios.
The discussion of the probit results needs to be enhanced. For example, why are women more likely to work than men?
Why are age and education excluded from the probit analysis?
Throughout the paper the author(s) needs to pay better attention to the difference between unemployed and out of the labor force
The author spends far too much time discussing independent sample tests and far too little time on the probit model.
Author Response
The authors' notes to Reviewer 1 is in the attached document.

Reviewer 2 Report
Review of “Determinants of young people with secondary education being employed” (Economies – 211577)
Thank you for the opportunity to review this manuscript. The authors administered a survey about employment status to individuals in Lithuania with secondary education aged 18 to 25. Survey results highlight a range of factors that are positively associated with the likelihood an individual is employed. Implications for the study
I was surprised by some of the most basic findings and wondered if they could be cross-validated with other data sources. For example, women were far more likely to be employed or to have ever held a job. Is this simply because young men are more likely to be in school? The authors do mention one other employment survey of Lithuania youth – is that the only other source of employment data in Lithuania? If so, then you might consider raising this as an important contribution of the manuscript. If not, then I would further describe that past work to see if it aligns with the survey data findings, especially around gender and the overall percentages employed (48%).
The manuscript would be improved if the authors were to expand on the implications of their study. Based on the results of the study, is there anything you would recommend that local, regional, or national governments in Lithuania could do to increase youth unemployment? For example, I wasn’t clear whether, broadly speaking the authors concluded that unemployed youth are primarily unemployed voluntarily or involuntarily. In the conclusion paragraph, the authors list out the five factors associated with employment. Some of these are more policy-malleable than others. The framing of that paragraph makes it sound like the chief issue with youth unemployment relates to convincing these folks to apply for a job (rather than convincing firms to hire them), suggesting they are mostly unemployed voluntarily. Is that right? Lastly, with respect to implications, are the findings, or some elements of them, generalizable beyond Lithuania?
Related to the above, I was unclear about the desired outcome. The authors note that youth unemployment can cause individual and economy-wide problems. But most survey respondents who are unemployed are not working because they are studying. Is that a concern from a policy perspective? I don’t have the answer, but I the authors should come down on one side of that question or the other.
The findings around contract type seemed over-emphasized (unless I missed something). Currently unemployed individuals are more likely to say their last (or current) job was temporary, while currently employed individuals are more likely to say their current (or last) job was permanent. This finding is not surprising, but it may suggest that currently unemployed youth who previously held a job left that position involuntarily. I wonder if that what this finding is meant to imply?
The paragraph on page 7 in lines 252 to 261 is confusing, e.g., the sentence “Both groups of young people evaluated working hours best” is confusing because unemployed people, by definition, are talking about their last job, while employed people, by definition, are talking about their current job. I think slowing down a little there and providing more explanation would be helpful.
The review of literature is an especially strong aspect of the manuscript.
Author Response
The authors' notes to Reviewer 2 is in the attached document.

Round 2
Reviewer 2 Report
Review of “Determinants of young people with secondary education being employed” (Economies – 211577.V2)
Thank you for the opportunity to review this revised manuscript. The authors administered a survey about employment status to individuals in Lithuania with secondary education aged 18 to 25. Survey results highlight a range of factors that are positively associated with the likelihood an individual is employed.
The authors have made several revisions that I believe have improved the manuscript. A few minor suggested edits remain:
The authors partially address a concern from my last review regarding the surprising finding that women youth have higher employment rates than male youth. I appreciate the authors have added further discussion of this finding and what it might mean. One suggestion I had was to explore whether this finding exists in other datasets in Lithuania, or if it may be true in other countries (or if other studies have reached similar findings). I wondered if the authors are aware of any other context in which employment patterns by gender follow similar patterns, with higher employment rates among females?
Some of the figures are somewhat difficult to read, such as the initial pie graph and the pie/ring graph comparing results by gender.
